# FAK/PYK2 promotes the Wnt/β-catenin pathway and intestinal tumorigenesis by phosphorylating GSK3β

Chenxi Gao[1,2†], Guangming Chen[1,2†], Shih-Fan Kuan[3], Dennis Han Zhang[4], David D Schlaepfer[5], Jing Hu[1,2*]

[1]Department of Pharmacology and Chemical Biology, University of Pittsburgh School of Medicine, Pittsburgh, United States; [2]University of Pittsburgh Cancer Institute, Pittsburgh, United States; [3]Department of Pathology, University of Pittsburgh School of Medicine, Pittsburgh, United States; [4]Dietrich School of Arts and Sciences, Pittsburgh, United States; [5]Department of Reproductive Medicine, Moores Cancer Center, University of California, San Diego, San Diego, United States

**Abstract** Aberrant activation of Wnt/β-catenin signaling plays an unequivocal role in colorectal cancer, but identification of effective Wnt inhibitors for use in cancer remains a tremendous challenge. New insights into the regulation of this pathway could reveal new therapeutic point of intervention, therefore are greatly needed. Here we report a novel FAK/PYK2/GSK3β$^{Y216}$/β-catenin regulation axis: FAK and PYK2, elevated in adenomas in *APC*$^{min/+}$ mice and in human colorectal cancer tissues, functioned redundantly to promote the Wnt/β-catenin pathway by phosphorylating GSK3β$^{Y216}$ to reinforce pathway output—β-catenin accumulation and intestinal tumorigenesis. We previously showed that Wnt-induced β-catenin accumulation requires Wnt-induced GSK3β/β-TrCP interaction; the current study revealed that phosphorylation of GSK3β$^{Y216}$ was a molecular determinant of GSK3β recruitment of β-TrCP. Pharmacological inhibition of FAK/PYK2 suppressed adenoma formation in *APC*$^{min/+}$ mice accompanied with reduced intestinal levels of phospho-GSK3β$^{Y216}$ and β-catenin, indicating that FAK/PYK2/GSK3β$^{Y216}$ axis is critical for the activation of Wnt/β-catenin signaling in APC driven intestinal tumorigenesis.

*For correspondence: huj3@upmc.edu

†These authors contributed equally to this work

Competing interests: The authors declare that no competing interests exist.

## Introduction

Aberrant activation of the canonical Wnt (Wnt/β-catenin) pathway is crucial for many cancer types, especially those of the gastrointestinal tract: genetic alterations in Wnt signaling occur in 93% of human CRCs, among which inactivation of tumor suppressor *adenomatous polyposis coli* (*APC*) (85%) and activating mutations of proto-oncogene *β-catenin* (5%) (*Cancer Genome Atlas Network, 2012*) are mutually exclusive (*Sparks et al., 1998*). The loss of functional APC results in less efficient GSK3 (two isoforms in mammals, GSK3α and GSK3β)-mediated phosphorylation of β-catenin, leading to reduced degradation of β-catenin, thus mimicking Wnt stimulation. Although *APC* mutation is believed to activate Wnt signaling constitutively, histopathological studies show that only about 50% colon carcinoma cells display homogeneous nuclear β-catenin staining (*Chung et al., 2001*), a surrogate for Wnt signaling activity, indicating that *APC* mutation alone is not sufficient to cause persistent or full activation of the Wnt pathway in CRC cells, therefore Wnt signaling is regulatable in *APC*-mutated CRCs. But how Wnt signaling is regulated in intestinal tumorigenesis remains largely undefined.

A key feature of the Wnt/β-catenin pathway is the tight control of the abundance of transcription co-activator β-catenin by the β-catenin destruction complex, consisting of GSK3, casein kinase 1 (CK1), APC, scaffold protein Axin and β-catenin. In Wnt-off state, the cytosolic level of β-catenin is kept low

**eLife digest** The cells in our body communicate with each other to coordinate many essential processes, including cell division and the formation of tissues and organs. The Wnt signaling pathway is crucial for cell communication across all animal species, but activating this pathway at the wrong time can cause cancer to develop. As part of the investigation into treatments for colon and other intestinal cancers, researchers have tried to identify drugs that inhibit Wnt signaling. This search would be easier if we understood more about how the Wnt signaling pathway is controlled.

When the protein GSK3 is active, it can switch the Wnt signaling pathway 'off', and inactivating GSK3 can turn the Wnt signaling pathway 'on'. Enzymes known as kinases can inactivate GKS3 by adding a phosphate group to it, in a process known as phosphorylation. It was unknown which kinases phosphorylate GSK3, and whether this affects how cancerous tumors develop in the colon.

Gao, Chen et al. analyzed cells taken from human and mice and observed that two kinases called PYK2 and FAK phosphorylate one form of GSK3, known as GSK3β. Furthermore, both kinases are required to inactivate GSK3β completely, and so turn on the Wnt signaling pathway.

During the early stages of colon cancer, cells first develop into polyps, which subsequently become cancerous. Gao, Chen et al. treated mice that had genetic mutations that made them susceptible to colon cancer with a chemical compound that inhibits both FAK and PYK2. These mice grew fewer polyps than untreated mice, and the polyps that did grow tended to be smaller.

Tissue samples taken from humans in the early stages of colon cancer—as the polyps progress towards becoming cancerous—had high levels of FAK, PYK2 and phosphorylated GSK3β. Overall, this suggests that drugs that simultaneously inhibit FAK and PYK2 may be an effective treatment for colon cancer, although further studies will be needed to confirm this.

by its continuous degradation: β-catenin is phosphorylated by GSK3 on serine 33 and 37 and threonine 41, phosphorylation then triggers β-catenin recruitment of ubiquitin E3 β-TrCP (β-transducin repeats-containing proteins), causing its ubiquitination and proteasomal degradation (*Wu and Pan, 2010*). In Wnt-on state, Wnt ligand forms a complex with the cell-surface receptor Frizzled and low-density lipoprotein receptor-related protein (LRP) 5/6 (*Wu and Pan, 2010*), and initiates a series of molecular events ultimately causing stabilization of β-catenin by suppressing phosphorylation of β-catenin (*Hernandez et al., 2012*; *Kim et al., 2013*) as well as β-TrCP-mediated ubiquitination and proteasomal degradation of β-catenin (*Li et al., 2012*). Newly synthesized β-catenin then accumulates and enters the nucleus to interact with transcription factors TCF (T-cell factor)/LEF (lymphoid enhancing factor) to activate transcription of the Wnt target genes (*Li et al., 2012*).

How GSK3 is modulated and fine-tuned by Wnt to achieve β-catenin stabilization remains a fundamental question in the field of Wnt signaling, with many puzzles and contradictions. We previously found that Wnt simulation induces monoubiquitination of GSK3β by β-TrCP, which is required for Wnt-induced inhibition of β-catenin recruitment of β-TrCP and subsequent stabilization of β-catenin (*Gao et al., 2014*). This study addressed important questions left unanswered: What are the molecular determinants of GSK3β recruitment of β-TrCP? β-TrCP is the substrate recruitment module of the SCF$^{β-TrCP}$ ubiquitin ligase supercomplex, recruitment of β-TrCP to its substrate, for example β-catenin, generally requires substrate phosphorylation (*Cardozo and Pagano, 2004*). If this is also true in GSK3β's case, what is the involved phosphorylation site(s)? What are the kinases responsible for catalyzing the phosphorylation? Is this phosphorylation involved in intestinal tumorigenesis? Our results indicate that phosphorylation of GSK3β$^{Y216}$ by FAK and PYK2 is a molecular determinant of GSK3β recruitment of β-TrCP, and is required for the activation of the Wnt/β-catenin pathway and crucial for Wnt-driven intestinal tumorigenesis.

## Results

### GSK3β phosphorylation at Y216 is required for GSK3β interaction with β-TrCP and subsequent ubiquitination

β-TrCP binds to its substrates through phosphorylated residues in conserved degradation motifs (DpSGxxpS) (*Cardozo and Pagano, 2004*), but tyrosine phosphorylation of substrate is also a recruiter

for β-TrCP (*Sahasrabuddhe et al., 2015*). GSK3β is known to be phosphorylated at serine 9, threonine 43, tyrosine 216 and threonine 390 (*Gao et al., 2014*). To determine which of these sites is involved in GSK3β recruitment of β-TrCP, we generated single point mutations at these residues and compared the interactions between the mutants and β-TrCP. We found that mutation of S9 to alanine (A) moderately inhibited GSK3β/β-TrCP interaction, whereas mutation of T43 and T390 to alanine had no impact on GSK3β affinity for β-TrCP (*Figure 1A*). In contrast, mutation of Y216 to phenylalanine (F) almost completely abrogated GSK3β binding to β-TrCP. Mimicking GSK3β$^{Y216}$ phosphorylation by mutation Y216 to glutamic acid (E) restored the interaction, suggesting that it was phosphorylation that determines the recruitment of β-TrCP to GSK3β.

Consistent with β-TrCP's role in GSK3β monoubiquitination (*Gao et al., 2014*), mutation of GSK3β Y216 to phenylalanine substantially reduced GSK3β ubiquitination in HEK293T cells (*Figure 1B*) and in an in vitro reconstituted system (*Figure 1C*, left panel). In the presence of SCF$^{β-TrCP1}$ ubiquitin E3 ligase complex, WT GSK3β and its phosphorylation mimicking mutant Y216E, but not Y216F mutant, were ubiquitinated. Of note, in vitro synthesized GSK3β through Rabbit Reticulocyte Lysate Translation Systems were weakly phosphorylated at Y216 and incubation with ATP further enhances GSK3β$^{Y216}$ phosphorylation through GSK3β autophosphorylation (*Cole et al., 2004*) (*Figure 1C*, right panel), which explains why in vitro ubiquitination of WT GSK3β was possible if GSK3β monoubiquitination requires Y216 phosphorylation. Although Wnt3a treatment did not alter p-GSK3β$^{Y216}$ level in HEK293T cell (*Figure 1D*) (GSK3α$^{Y279}$ corresponds to GSK3β$^{Y216}$, the antibody recognizing p-GSK3β$^{Y216}$ also recognizes p-GSK3α$^{Y279}$), the loss of GSK3β$^{Y216}$ phosphorylation greatly impaired both basal and Wnt3a-induced GSK3β monoubiquitination (*Figure 1E*).

Prior studies indicate that phosphorylation of GSK3β$^{Y216}$ may or may not affect GSK3β kinase activity and GSK3β/Axin interaction (*Hughes et al., 1993*; *Zhang et al., 2001*; *Dajani et al., 2003*; *Buescher and Phiel, 2010*). The current study showed that GSK3β$^{Y216F}$ mutant protein interacted with Axin1 as strongly as WT GSK3β in HEK293T cells (*Figure 1F*). Same as GSK3β ubiquitination mutant GSK3β$^{KKKK15/27/32/36RRRR}$ (lysines 15/27/32/36 to arginines mutant, labeled as S4 in the figure), GSK3β$^{Y216F}$ and GSK3β$^{Y216E}$ phosphorylated β-catenin as efficiently as WT GSK3β in vitro (*Figure 1G*, upper panel) and in cells (*Figure 1G*, lower panel). Together, these results imply that phosphorylation status on Y216 neither affects GSK3β binding affinity for Axin1 nor changes GSK3β kinase activity towards β-catenin phosphorylation.

## Phosphorylation of GSK3β$^{Y216}$ is required for the activation of Wnt/β-catenin signaling

We next validated the role of GSK3β$^{Y216}$ in Wnt/β-catenin signaling in GSK3β knockout (KO) MEFs (mouse embryonic fibroblasts). GSK3α and GSK3β are functionally redundant in Wnt signaling (*Doble et al., 2007*), to avoid compensation from GSK3α, we generated GSK3β KO/GSK3α KD MEF cell line. Loss of both GSK3α/β increased the basal level of β-catenin (*Figure 2A*, comparing lanes 1 and 5), presumably due to the lack of GSK3-mediated phosphorylation of β-catenin$^{S33/S37/T41}$ (*Doble et al., 2007*). Reconstitution of the GSK3β KO/GSK3α KD MEFs with WT HA-GSK3β reduced the basal level of β-catenin to the level comparable to WT MEFs (*Figure 2A*, comparing lanes 1, 5 and 7), likely due to the restoration of GSK3-mediated phosphorylation of β-catenin. Reconstitution these MEFs with WT HA-GSK3β and its phosphorylation mimetic mutant (HA-GSK3β$^{Y216E}$), but not GSK3β$^{Y216F}$ mutant, also restored Wnt-induced accumulation of β-catenin, thus validating the positive role of GSK3β$^{Y216}$ phosphorylation in Wnt signaling. The fact that GSK3β$^{KKKK15/27/32/36RRRR/Y216E}$ mutant failed to restore Wnt-induced β-catenin stabilization (*Figure 2A*, lanes 13 and 14) functionally confirmed that Y216 phosphorylation occurs prior to ubiquitination. We have found that Wnt-induced long-term β-catenin stabilization is mediated by GSK3β monoubiquitination through suppressing β-catenin recruitment of β-TrCP (*Gao et al., 2014*). In line with this, the co-IP results showed that upon Wnt3a stimulation, β-TrCP recruitment to β-catenin and ubiquitination of the phosphorylated β-catenin$^{S33/S37/T41}$ (*Figure 2B*, left panel) were substantially increased in GSK3β KO/GSK3α KD MEFs reconstituted with HA-GSK3β$^{Y216F}$, whereas reconstitution with Y216 phosphorylation mimetic mutant (HA-GSK3β$^{Y216E}$) exerted the opposite effect. The results from the reciprocal IP assay confirmed more β-TrCP-bound p-β-catenin$^{S33/S37/T41}$ in HA-GSK3β$^{Y216F}$ cells (*Figure 2B*, right panel). Together, these data suggested that phosphorylation of GSK3β$^{Y216}$ is required for the activation of the Wnt/β-catenin signaling by suppressing phosphorylated β-catenin$^{S33/S37/T41}$ recruitment of β-TrCP.

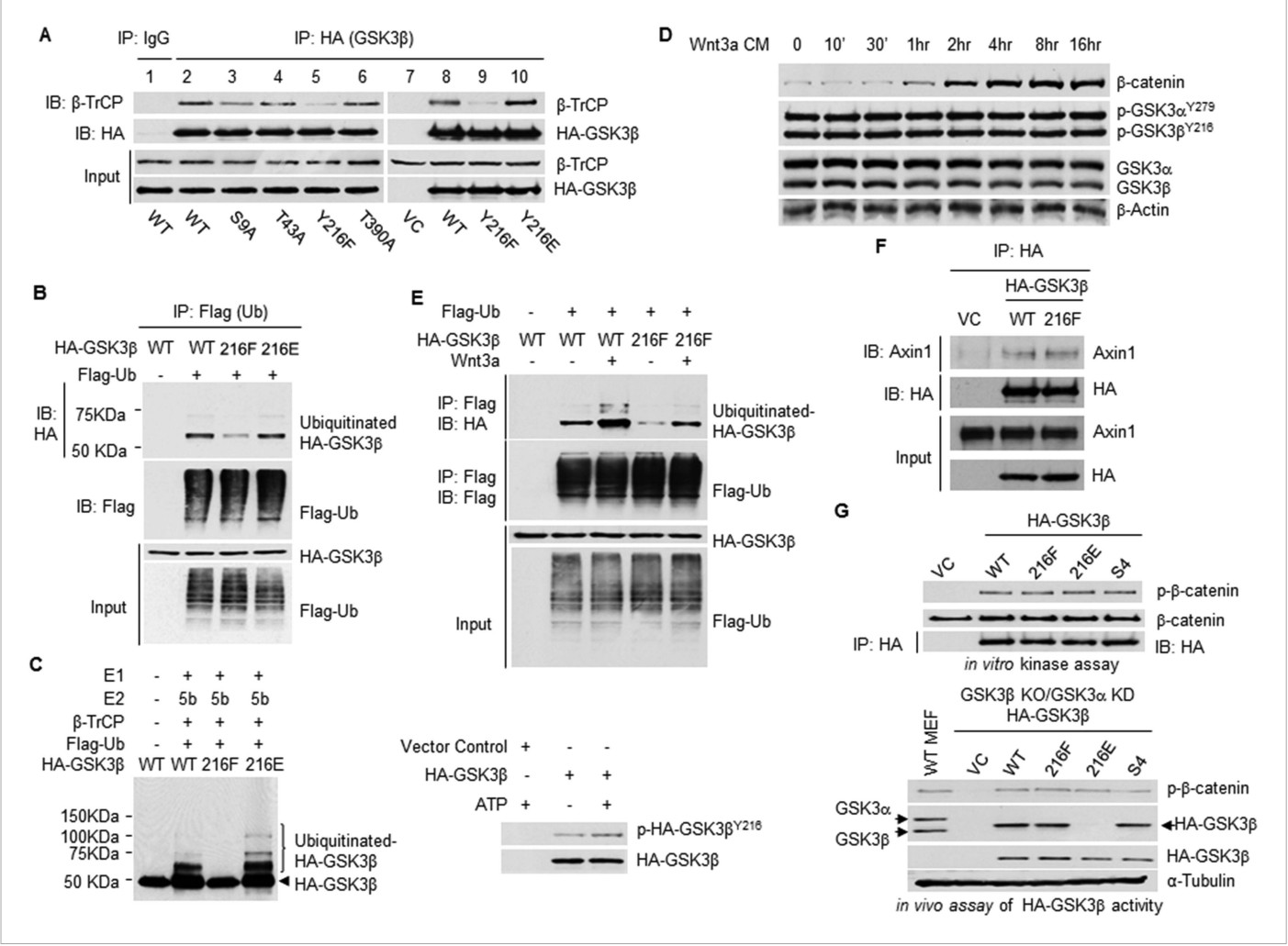

**Figure 1**. β-TrCP-mediated GSK3β ubiquitination requires GSK3β$^{Y216}$ phosphorylation. (**A**) The whole cells lysates from HEK293T cells transfected with empty vector or constructs expressing HA-tagged GSK3β or its phosphorylation mutants were used for immunoprecipitation (IP) and immunoblotting (IB) analysis with indicated antibodies. (**B**) Ubiquitination of HA-GSK3β was evaluated by IP with anti-FLAG using whole cell lysates derived from HEK293T cells transfected with indicated constructs followed by IB analysis with anti-HA. (**C**) In vitro synthesized wild type (WT) and mutant HA-GSK3β proteins were immunoprecipitated with anti-HA affinity gel. Left panel: in vitro ubiquitination assay was performed by incubating immunoprecipitated HA-GSK3β with E1, E2, Ub and SCF$^{β-TrCP1}$ in the presence of ATP at 37°C for 2 hr. The reaction mixtures were analyzed by IB. Right panel: the immunoprecipitated HA-GSK3β was incubated with ATP at 37°C for 2 hr before subjecting to IB using antibody recognizing phosphorylated GSK3β$^{Y216}$. (**D**) HEK293T cells were treated with Wnt3a-conditioned medium (CM) for the indicated time before harvesting for IB analysis. (**E**) HEK293T cells were transfected with indicated plasmids and treated with control-CM or Wnt3a-CM for 6 hr. Ubiquitination of HA-GSK3β WT and Y216F mutant protein were evaluated by IP with anti-FLAG followed by IB with anti-HA. (**F**) HEK293T cells were transfected with constructs expressing HA-GSK3β or its mutant. 24 hr post-transfection, the cells were harvested for IP with anti-HA followed by IB with anti-Axin1. (**G**) Upper panel: HA-tagged GSK3β or its mutants were ectopically expressed in HEK293T cells and immunoprecipitated with anti-HA. The precipitated proteins were incubated with purified CK1 and β-catenin in the presence of ATP at 37°C for 1 hr. Kinase activity was evaluated by IB with antibody recognizing phosphorylation of β-catenin$^{S33/S37/T41}$ catalyzed by GSK3β. Lower panel: the whole cell lysates from WT MEFs, GSK3β knockout (KO) and GSK3α knockdown (KD) MEFs reconstituted with empty vector or indicated HA-GSK3β mutants were used for IB. Antibody recognizing both GSK3α and GSK3β was used for detecting endogenous GSK3 and HA-tagged GSK3β. This antibody recognized HA-GSK3β$^{Y216F}$ but not HA-GSK3β$^{Y216E}$ mutant protein.

CRC cell lines including SW480 (APC-deficient, WT β-catenin) are Wnt autocrine cell lines (*Bafico et al., 2004*). In SW480 cells, KD of both GSK3α and GSK3β greatly inhibited cell growth in soft agar (*Figure 2C*) and in nude mice (*Figure 2D*). Reconstitution of the cells with WT HA-GSK3β or HA-GSK3β$^{Y216E}$ but not HA-GSK3β$^{Y216F}$ restored cell growth in vitro and in vivo, implying that phosphorylation of GSK3β$^{Y216}$ is functionally important in CRC cells.

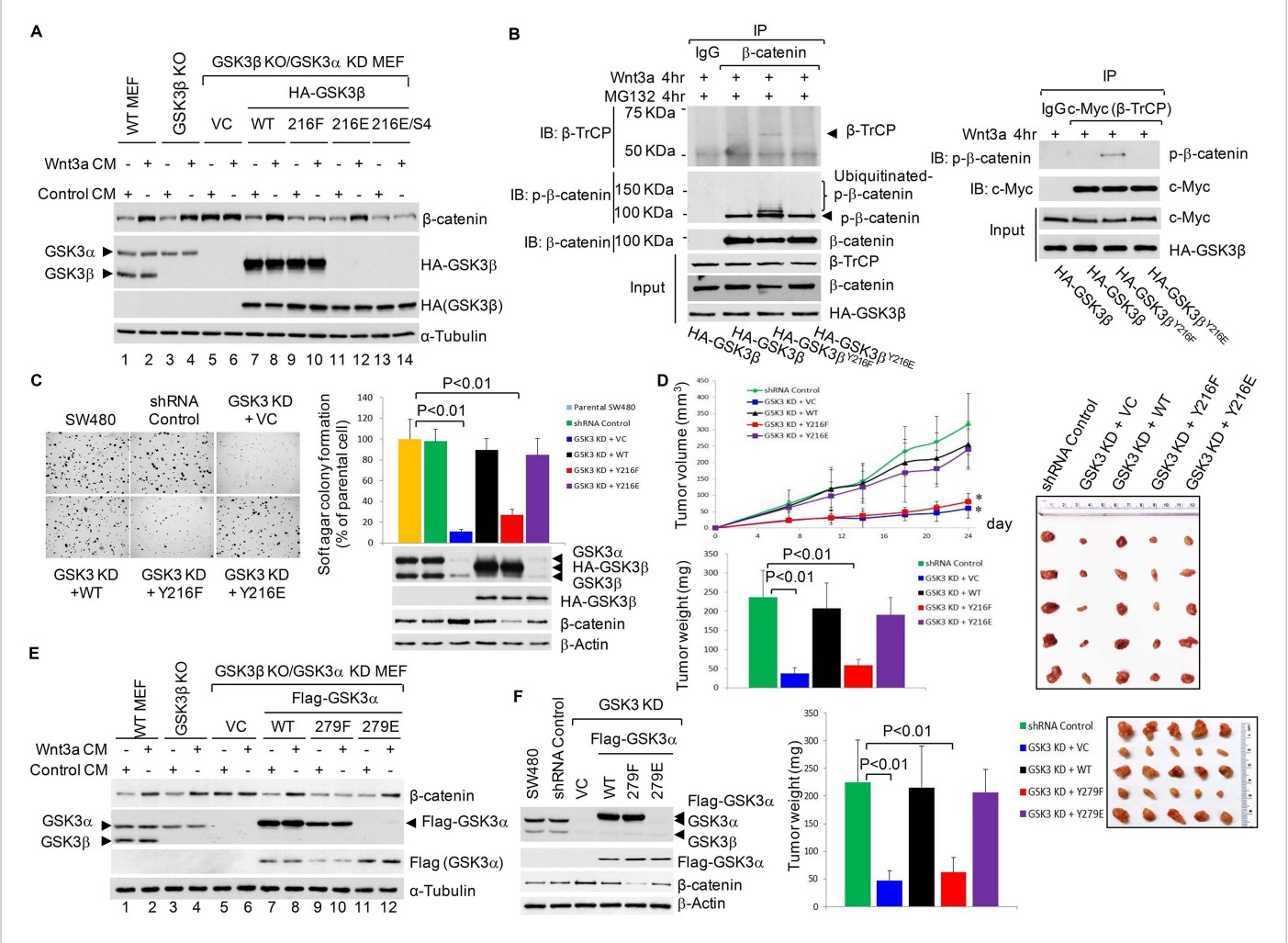

**Figure 2**. Activation of Wnt/β-Catenin signaling requires GSK3β$^{Y216}$ phosphorylation. (**A**) IB analysis using whole cell lysates from indicated MEFs treated with control-CM or Wnt3a-CM for 6 hr. Anti-GSK3 was used to detect endogenous GSK3 and HA-GSK3β. This antibody recognized HA-GSK3β$^{Y216F}$ but not HA-GSK3β$^{Y216E}$ or GSK3β$^{Y216E/S4}$ mutant proteins. (**B**) The GSK3β KO/GSK3α KD MEFs were transfected with vectors expressing indicated proteins, treated with Wnt3a-CM and proteasome inhibitor MG132 (10 μM) for 4 hr as indicated, and then harvested for IP and IB analyses using indicated antibodies. (**C**) Left panel, SW480 cells and GSK3α/β KD SW480 cell lines expressing vector control, shRNA-resistant WT HA-GSK3β or its mutants were used for soft agar assays. Pictures were taken 16 days after plating at 40×. Right panel, quantification of soft agar assays. Data are presented as mean ±SD. Expressions of GSK3α/β (endogenous or mutants) and β-catenin were evaluated by IB at the time of plating. (**D**) Upper left panel, SW480 cell lines used in (**C**) were injected into both flanks of nude mice (5 mice in each cell group). Tumor sizes were measured at indicated time points. Tumor volumes were calculated and plotted. * denotes a statistically significant difference (p < 0.01) comparing with shRNA control group. Bottom left panel, tumor weighs from different groups at the end of the experiment. Values are means ±SD (n = 10). Right panel, representative tumors from different groups. (**E**) IB analysis using whole cell lysates from indicated MEFs treated with control-CM or Wnt3a-CM for 6 hr. Anti-GSK3 was used to detect endogenous GSK3 and Flag-GSK3α. This antibody recognized Flag-GSK3α$^{Y279F}$ but not Flag-GSK3α$^{Y279E}$ mutant protein. (**F**) Left panel: IB characterization of the SW480 cell lines. Middle panel: indicated SW480 cell lines were injected into both flanks of nude mice (5 mice in each cell group). Tumors from different groups were weighed at the end of the experiment. Values are means ±SD (n = 10). * denotes a statistically significant difference (p < 0.01) comparing with shRNA control group. Right panel, representative tumors from different groups.

GSK3α$^{Y279}$ corresponds to GSK3β$^{Y216}$. Given the redundant role of GSK3α and GSK3β in Wnt signaling (*Doble et al., 2007*), we next examined whether phosphorylated GSK3α$^{Y279}$ and GSK3β$^{Y216}$ also function redundantly in Wnt signaling and in CRC cells. The results showed that reconstitution of the GSK3β KO/GSK3α KD MEFs with WT Flag-GSK3α or Flag-GSK3α$^{Y279E}$, but not GSK3α$^{Y279F}$ mutant restored Wnt-induced accumulation of β-catenin (*Figure 2E*). In GSK3α/β KD SW480 cells, reconstitution with WT Flag-GSK3α or Flag-GSK3α$^{Y279E}$ but not Flag-GSK3α$^{Y279F}$ restored in vivo

tumor growth in a cell line xenograft assay (*Figure 2F*). Together, these data indicated that GSK3α$^{Y279}$ and GSK3β$^{Y216}$ function redundantly in the Wnt/β-catenin pathway and in CRC cells.

## FAK/PYK2 phosphorylate GSK3β$^{Y216}$

To identify the kinases responsible for GSK3β$^{Y216}$ phosphorylation, we performed a pilot screening experiment using various kinase inhibitors. In line with a prior report that GSK3β$^{Y216}$ can be phosphorylated by PYK2 (*Sayas et al., 2006*), we found that treatment of HEK293T cells with PF-562271, a dual inhibitor of FAK/PYK2, inhibited GSK3β$^{Y216}$ and GSK3α$^{Y279}$ phosphorylation in the presence and absence of Wnt3a (*Figure 3A*). FAK and PYK2 are closely related non-receptor tyrosine kinases, participating in diverse signaling pathways and control a number of cellular processes (*Mitra et al., 2005*; *Schaller, 2010*). Unlike ubiquitously expressed FAK, PYK2 has a more restricted tissue expression and is highly enriched in the brain and hematopoietic cells (*Avraham et al., 2000*). Our IB results (*Figure 3B*) showed that while HEK293T and HT-29 cells predominantly expressed FAK, SW480 cells primarily expressed PYK2. Both FAK and PYK2 were expressed in DLD-1 cells. In HEK293T cells, KD of FAK abrogated GSK3α$^{Y279}$ and GSK3β$^{Y216}$ phosphorylation and Wnt-induced β-catenin accumulation (*Figure 3C*). In CRC cells predominantly expressing either FAK (HT-29 cells) or PYK2 (SW480 cells), KD of FAK or PYK2 alone was sufficient to reduce the levels of phosphorylated GSK3α$^{Y279}$ and GSK3β$^{Y216}$ and total β-catenin (*Figure 3D,E*). However, in DLD-1 cells expressing FAK and PYK2, simultaneous KD of both kinases were required to inhibit GSK3$^{Y279/Y216}$ phosphorylation and to reduce the level of β-catenin (*Figure 3F*), strongly suggesting that FAK and PYK2 function redundantly to phosphorylate GSK3$^{Y279/Y216}$ to promote Wnt/β-catenin signaling. To further confirm that GSK3 is a phosphorylation substrate of FAK/PYK2, we examined whether FAK/PYK2 binds to GSK3β and FAK/PYK2's ability to phosphorylate GSK3β$^{Y216}$ in vitro. In line with prior findings that GSK3β interacts with FAK and PYK2 in vivo (*Huang et al., 2002*; *Sayas et al., 2006*), our co-IP results showed that GSK3β interacted with PYK2 in SW480 cells and with FAK in HT-29 cells (*Figure 3G*). However, whether they bind directly or indirectly is not known. The result of in vitro kinase assay confirmed that FAK and PYK2 were able to directly phosphorylate GSK3β$^{Y216}$ in vitro (*Figure 3H*). Together, these results reveal that FAK and PYK2 mediate GSK3$^{Y279/Y216}$ phosphorylation in CRC cells.

## FAK/PYK2 kinase activity mediates FAK/PYK2 regulation of the Wnt/β-catenin pathway

The above results identified GSK3β as a new substrate of FAK kinase. FAK has kinase-dependent and kinase-independent scaffolding activities (*Cance et al., 2013*). If FAK/PYk2 regulates Wnt/β-catenin through their kinase activity by phosphorylating GSK3β$^{Y216}$, there should be differences in the time courses of FAK/PYK2 inhibitor-mediated suppression of FAK/PYK2 kinase activity, GSK3β$^{Y216}$ phosphorylation and β-catenin. The results confirmed that treatment with PF-562271 caused sequential inhibition of PYK2 kinase activity represented as phosphorylated PYK2$^{Y402}$ (*Park et al., 2004*), GSK3β$^{Y216}$ phosphorylation and β-catenin (*Figure 4A*). To ultimately validate that it is the kinase activity that mediates FAK/PYK2 regulation of Wnt/β-catenin signaling, we examined GSK3β$^{Y216}$ phosphorylation-dependent activation of Wnt/β-catenin signaling in FAK kinase-deficient MEFs (FAK$^{R454/R454}$ knockin MEFs). A knock-in point mutation of lysine 454 to arginine within the catalytic domain inactivates FAK kinases activity (*Lim et al., 2010*) (*Figure 4B*, represented by abolished phosphorylation of FAK$^{Y397}$) but leaves FAK's scaffolding activity intact (*Lim et al., 2008*). Consistent with our finding that FAK and PYK2 function redundantly in phosphorylating GSK3, despite of suppressed FAK activity in FAK$^{R454/R454}$ MEFs, KD of PYK2 was required to inhibit GSK3β$^{Y216}$ phosphorylation. shRNA KD of PYK2 also abrogated Wnt3a-induced β-catenin accumulation; whereas overexpression of GSKβ$^{Y216E}$ restored Wnt3a-indcued β-catenin stabilization in the FAK$^{R454/R454}$/PYK2 KD MEFs (*Figure 4C*), implying that FAK/PYK2 stabilize β-catenin through phosphorylating GSKβ$^{Y216}$. We further found that β-TrCP recruitment to and ubiquitination of the phosphorylated β-catenin$^{S33/S37/T41}$ were greatly increased in FAK$^{R454/R454}$/PYK2 KD MEFs, ectopic expression of GSKβ$^{Y216E}$ exerted the opposite effect (*Figure 4D*). Together, these results indicated that FAK/PYK2 stabilizes β-catenin through FAK/PYK2 kinase-mediated phosphorylation of GSKβ$^{Y216}$. In line with the observation that elevated p-GSK3β$^{Y216}$ was significantly higher in 5FU treatment-resistant CRC patients than in 5FU-responsive patients (*Grassilli et al., 2013*), we also found that overexpression of GSKβ$^{Y216E}$ substantially prevented PF-562271 treatment-mediated inhibition of cell proliferation in SW480 cells (*Figure 4E*).

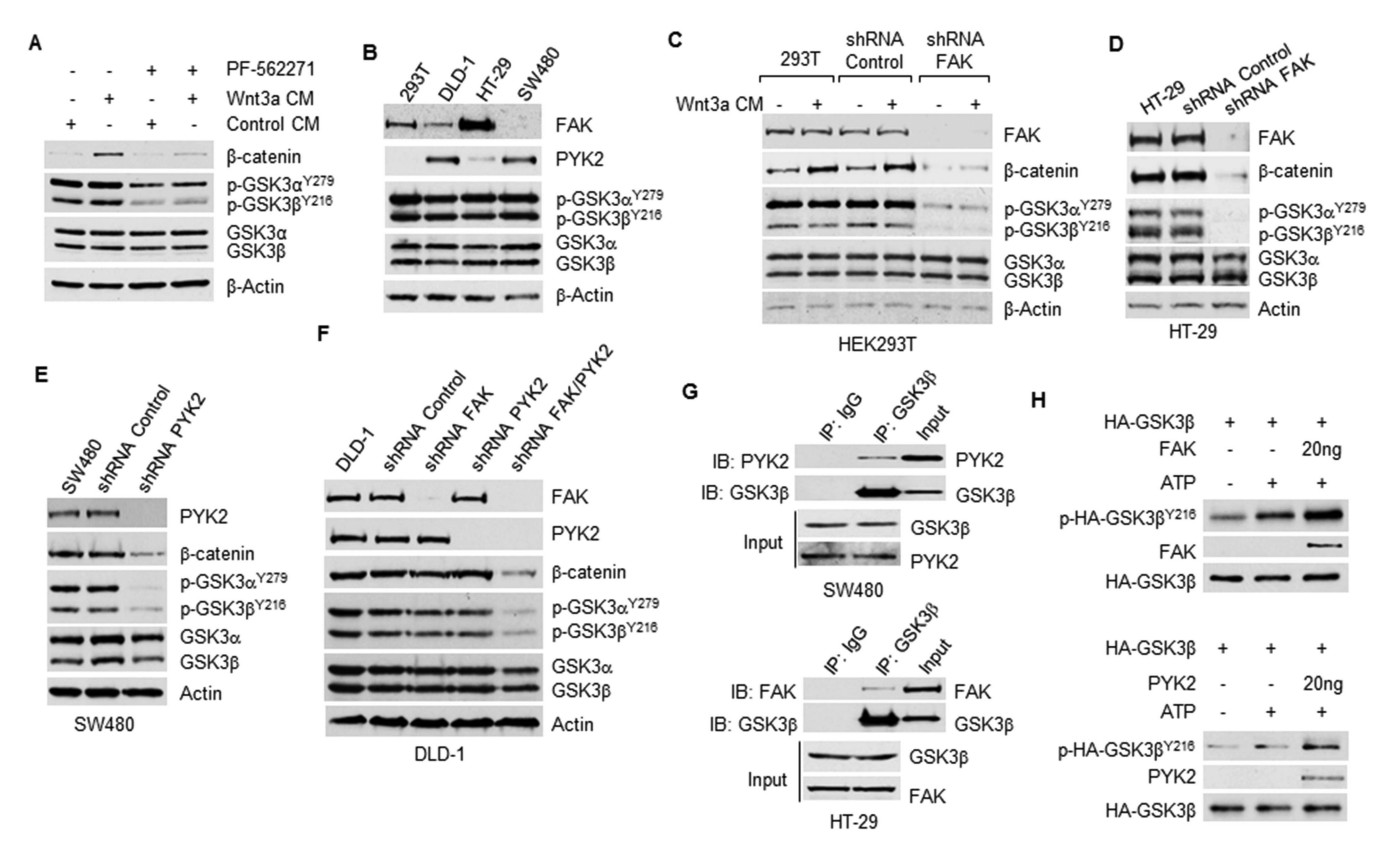

**Figure 3**. Phosphorylation of GSK3β$^{Y216}$ by FAK/PYK2. (**A**) HEK293T cells were pre-treated with or without PF-562271 (4 μM) for 4 hr followed by incubation with control-CM or Wnt3a-CM containing PF-562271 (4 μM) for additional 4 hr as indicated. The whole cell lysates were subjected for IB with indicated antibodies. (**B**) FAK and PYK2 expressions were evaluated by IB analysis. (**C**) Parental, FAK KD and shRNA control HEK293T cells were treated with control-CM or Wnt3a-CM for 4 hr before harvesting for IB analysis using indicated antibodies. (**D–F**) FAK KD, PYK2 KD or FAK/PYK2 KD cell lines and shRNA control cell lines were generated using lentiviral-based shRNA targeting FAK or PYK2 or control shRNA. The whole cell lysates were used to evaluate the levels of p-GSK3β$^{Y216}$ and β-catenin by IB with indicated antibodies. (**G**) GSK3β in SW480 and HT29 cells was immunoprecipitated with anti-GSK3β followed by IB with anti-PYK2 and anti-FAK, respectively. Normal IgG was used as IP control. (**H**) In vitro synthesized HA-GSK3β was immunoprecipitated with anti-HA. Beads-bound HA-GSK3β was then incubated with purified FAK or PYK2 recombined protein (or kinase buffer as a negative control) in the presence of ATP at 37°C for 2 hr. The reaction mixtures were subjected to IB analysis using indicated antibodies.

FAK can directly phosphorylate β-catenin at tyrosine 142 (*Chen et al., 2012*). We next examined whether PF-562271 treatment inhibits tumor growth when Y142 in β-catenin is mutated. The results from xenograft experiments showed that although overexpression of β-catenin$^{Y142E}$ partially jeopardized the anti-tumor effect of PF562271, PF562271 treatment significantly inhibited in vivo growth of SW480 cells ectopically expressing Flag-β-catenin$^{Y142E}$ (*Figure 4F*). While this result further proved the importance of FAK/PYK2-mediated GSK3 phosphorylation, it also suggested that phosphorylation of β-catenin$^{Y142}$ plays a role in FAK-targeted therapy for CRC.

## Inhibition of FAK/PYK2 suppresses intestinal phosphorylation of GSK3β$^{Y216}$ and adenoma formation in APC$^{min/+}$ mice

Intestine-specific deletion of FAK almost completely suppressed intestinal adenoma formation in *APC*-mutant mouse model for colorectal cancer (*Ashton et al., 2010*), proving that FAK is required for the oncogenicity of *APC* mutation in intestinal tumorigenesis. But the underlying mechanisms remain unclear. To determine whether FAK is involved in APC-driven tumorigenesis through its catalytic activity, we examined the anti-tumorigenic effects of dual FAK/PYK2 kinase inhibitor PF-562271 in *APC*$^{min/+}$ mice, a genetic model predisposed to the formation of intestinal adenomas due to aberrant

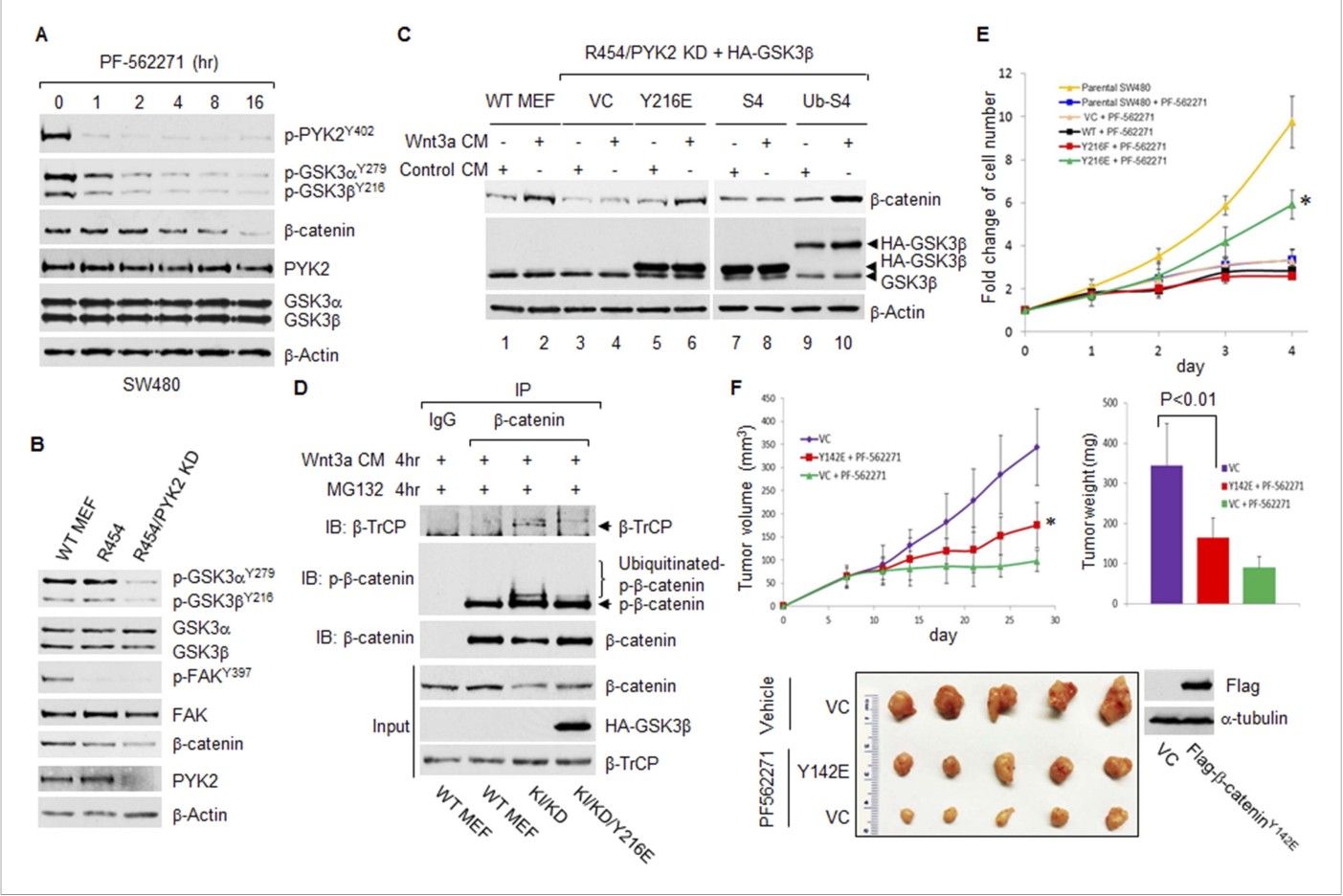

**Figure 4**. FAK/PYK2 promotes the Wnt/β-catenin pathway through phosphorylating GSK3βY216. (**A**) Time courses of PF-562271 treatment-induced inhibition of phosphorylation of PYK2, GSK3β and β-catenin in SW480 cells. The whole cell lysates were used for the IB analysis using indicated antibodies. (**B**) The whole cell lysates from indicated cells were used for IB analysis. (**C**) The WT MEFs, FAK^R454/R454/PYK2 KD MEFs transfected with empty vector or vector expressing indicated GSK3β mutants were treated with Wnt-3a CM or control-CM for 6 hr before harvesting for IB analysis. (**D**) The WT MEFs and FAK^R454/R454/PYK2 KD MEFs transfected with empty vector or vector expressing HA-GSK3β^Y216E were treated with Wnt3a-CM and proteasome inhibitor MG132 for 4 hr and then harvested for IP and IB analyses. KI/KD: FAK^R454/R454/PYK2 KD MEFs expressing empty vector. KI/KD/Y216E: FAK^R454/R454/PYK2 KD MEFs expressing HA-GSK3β^Y216E. (**E**) Parental SW480 cells and SW480 cell lines expressing vector control, HA-GSK3β or its mutants were treated with 2 μM PF-562271 for 4 days. Cell viability was determined by CCK-8 assay, using SW480 parental cell treated with DMSO as control. Data are presented as mean ±SD. * denotes a statistically significant difference (p < 0.01) between PF-562271-treated parental SW480 cells and cells overexpressing HA-GSK3β^Y216E. (**F**) Upper left panel: indicated HT-29 cell lines were injected into both flanks of nude mice (5 mice in each cell group). Animals were treated with either vehicle (5% Gelucire) or PF-562271 (33 mg/kg in vehicle) as indicated by oral gavage twice daily for 3 weeks. Tumor sizes were measured at indicated time points. Tumor volumes were calculated and plotted. * denotes a statistically significant difference (p < 0.01) comparing with vehicle-treated group. Upper right panel, tumor weighs from different groups at the end of the experiment. Values are means ±SD (n = 10). Bottom panel, representative tumors from different groups.

activation of the Wnt/β-catenin signaling. We found that PF-562271 treatment significantly inhibited both polyp formation (number) and polyp expansion (size) (*Figure 5A,B*). The results of IB confirmed that loss of functional APC upregulated intestinal expression of c-Myc (β-catenin target gene) and FAK (c-Myc target) (*Figure 5C*, left panel). PF-562271 treatment inhibited intestinal FAK kinase activity, evidenced by reduced level of p-FAK^Y397 (*Avraham et al., 2000*), and remarkably reduced β-catenin level, accompanied by reduced expressions of c-Myc and FAK, consistent with the notion that FAK functions as both 'effector' and 'regulator' of β-catenin. In line with the previous report (*Carothers et al., 2006*), we also found that intestinal PYK2 expression was substantially increased in tumors of APC^min/+ mice (*Figure 5C*, right panel), PF-562271 treatment abrogated the induction (*Figure 5C*, left panel).

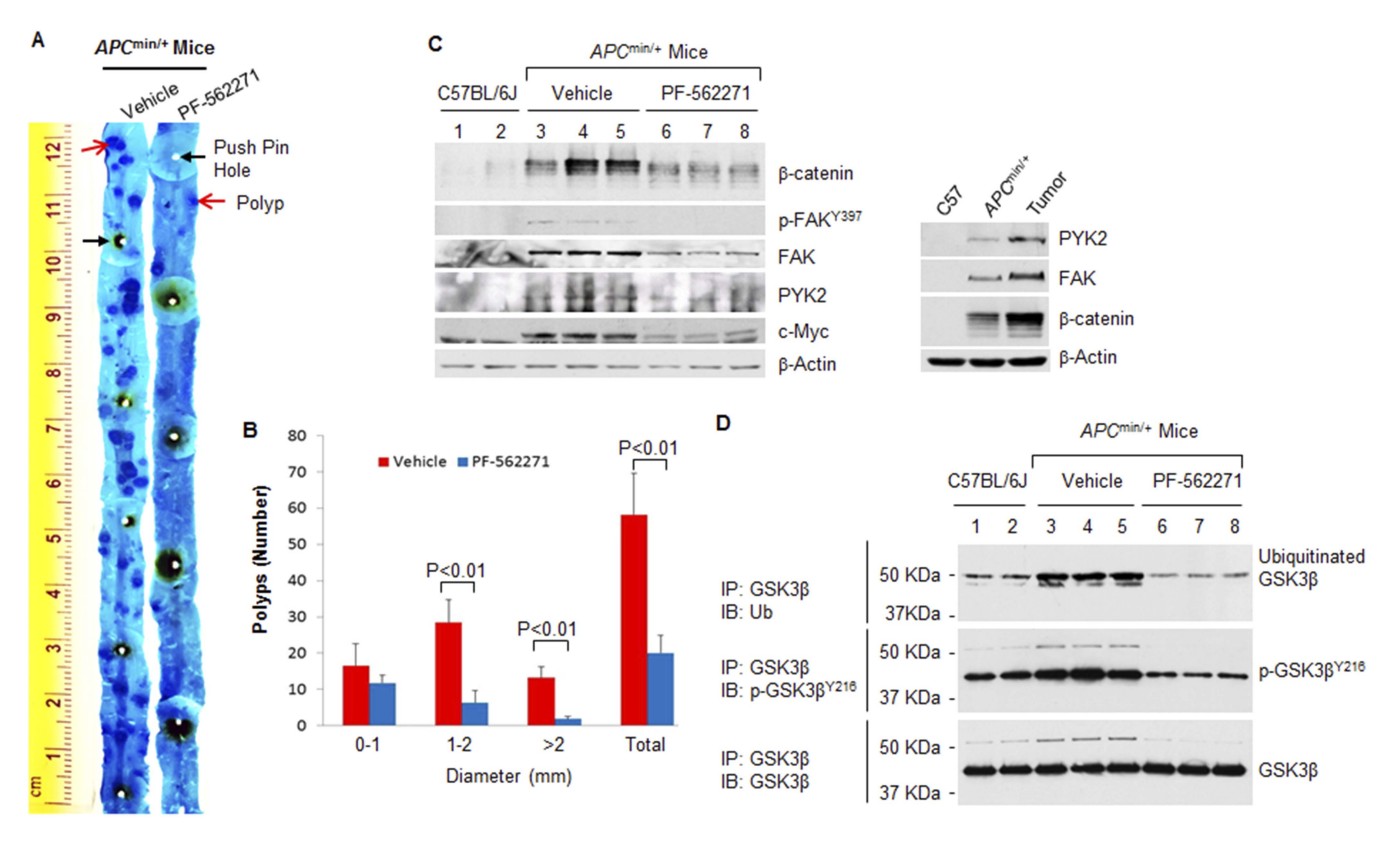

**Figure 5**. Inhibition of FAK/PYK2 kinase activity suppresses adenoma formation in $APC^{min/+}$ mice. (**A**) Representative staining results of ileum (distal) segments from $APC^{min/+}$ mice treated with vehicle or PF-562271 were shown. (**B**) Adenoma formation on entire small intestine was quantitatively assessed in $APC^{min/+}$ mice treated with vehicle (n = 5) or PF-562271 (n = 5). Data was presented as mean ±S.D. (**C**) Left panel: tissue lysates extracted from the scraped intestinal mucosa from indicated mice were used for IB analysis. Each lane represented a single mouse. Right panel: tissue lysates extracted from C57 control mouse, $APC^{min/+}$ mouse and polyps were used for IB. (**D**) Tissue lysates used in (**C**, left panel) were immunoprecipitated with anti-GSK3β. The beads-bound immunoprecipitates were resolved by SDS-PAGE and probed with indicated antibodies.

When assessing intestinal GSK3β$^{Y216}$ phosphorylation by IB, we detected multiple bands at high molecular weight that might represent phosphorylated GSK3α$^{Y279}$ and/or phosphorylated GSK3β$^{Y216}$ with other types of modifications, possibly ubiquitination. Unable to discriminate these possibilities by straight western blotting analysis, we chose to immunoprecipitate intestinal GSK3β using anti-GSK3β and then resolved the immunoprecipitates on SDS-PAGE gels, followed by IB analysis with anti-ubiquitin antibody and antibody recognizing phosphorylated GSK3β$^{Y216}$. In contrast to that endogenous ubiquitinated GSK3β in non-Wnt-treated resting cells was barely detectable in cell culture (*Gao et al., 2014*), to our surprise, substantial amount of the ubiquitinated intestinal GSK3β was readily detected in both C57BL/6J mice and $APC^{min/+}$ mice with much higher levels observed in $APC^{min/+}$ mice (*Figure 5D*). This finding highlights the importance of studying the role of posttranslational modifications in relevant biological contexts. The intestinal level of p-GSK3β$^{Y216}$ was also much higher in $APC^{min/+}$ mice than in control mice, whereas PF-562271 treatment greatly suppressed GSK3β$^{Y216}$ phosphorylation and GSK3β ubiquitination. Overall, these results strongly support that FAK/PYK2 kinase contributes to APC-driven intestinal tumorigenesis through phosphorylating GSK3β$^{Y216}$ to promote Wnt/β-catenin signaling.

## Positive correlation between the levels of FAK, PYK2, GSK3β, p-GSK3β$^{Y216}$ and β-catenin in human colorectal cancer tissues

Most colorectal carcinomas develop from adenomas via adenoma-carcinoma-sequence driven by the accumulation of genetic and epigenetic mutations (*Fearon and Vogelstein, 1990*); Our results

(*Figure 5*) imply that FAK/PYK2/GSK3β$^{Y216}$/β-catenin regulation axis is an early participant and driving force for intestinal tumorigenesis in *APC*$^{min/+}$ mice. If this is also true in the development of human CRC, dysregulation of FAK/PYK2 expression, GSK3β$^{Y216}$ phosphorylation and β-catenin accumulation should occur at the adenoma stage of the adenoma–carcinoma-sequence. In a proof-of-concept experiment, we analyzed serial sections of colon specimen from the same tissue block. We found that the IHC staining intensities of FAK, PYK2, GSK3β, p-GSK3α/β$^{Y279/Y216}$ and β-catenin were increased during the course of the normal-adenoma–carcinoma progression (*Figure 6A*), demonstrating that induction of FAK/PYK2 expression, GSK3α/β$^{Y279/Y216}$ phosphorylation and β-catenin accumulation did occur at the early stage (the adenoma stage) of colon cancer development. The positive correlation between these elevated expressions also suggested that these inductions were related events.

To determine whether the positive correlations between the levels of FAK, PYK2, p-GSK3α/β$^{Y279/Y216}$ and β-catenin are commonly seen in human CRC tissues, we performed IHC for these markers on CRC tissue microarray containing 34 cases of colorectal adenocarcinoma with 26 matched and 8 unmatched adjacent normal tissues. The results showed that the levels of FAK, PYK2, GSK3β, p-GSK3α/β$^{Y279/Y216}$ and β-catenin were all significantly elevated in tumor tissues, with more than 1 score point difference between normal and tumor tissue on average (*Figure 6B*). In the vast majority of samples,

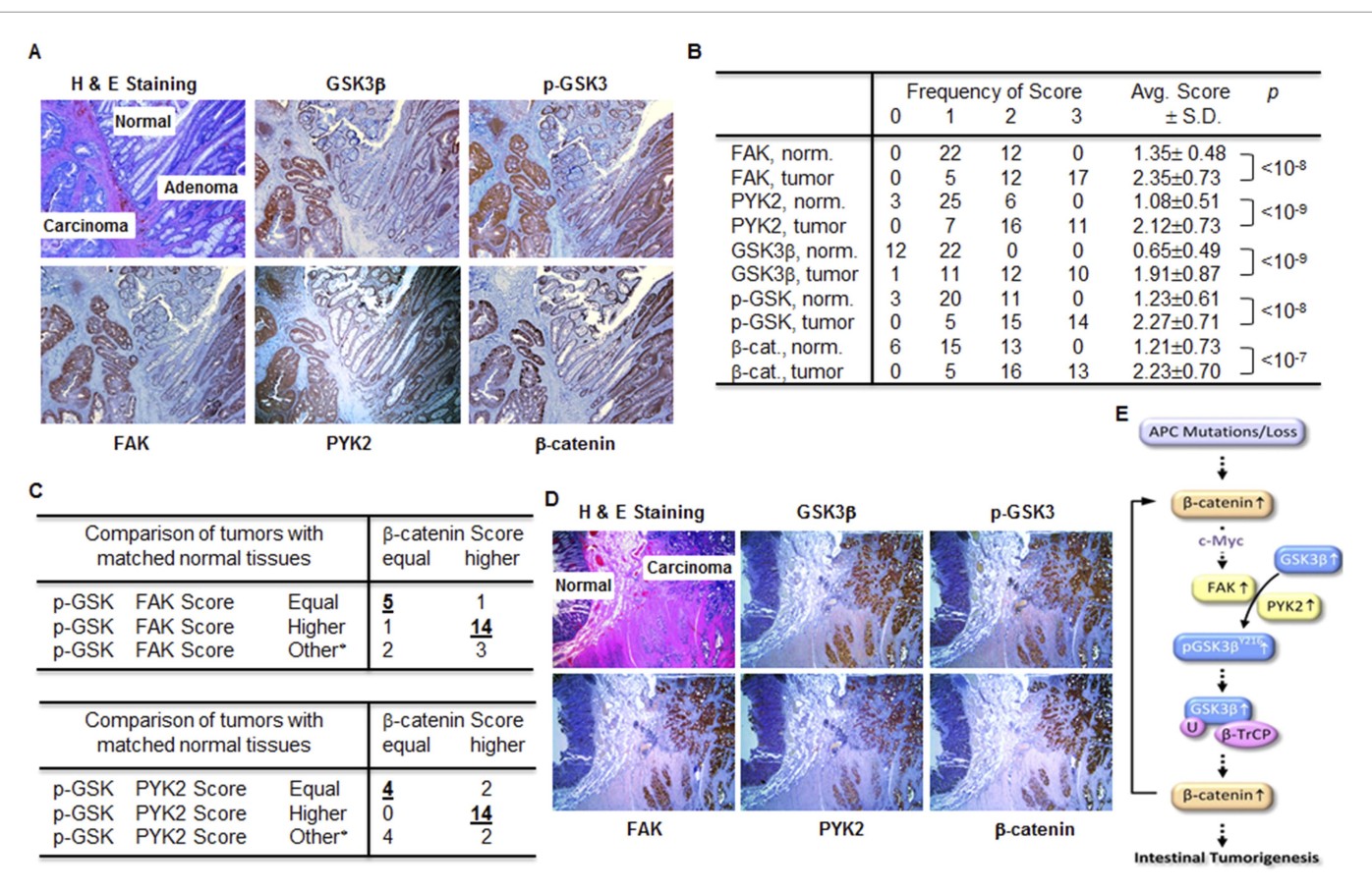

**Figure 6**. FAK, PYK2, p-GSK3β$^{Y216}$ and β-catenin positively correlate in human colorectal cancer tissues. (**A**) Hematoxylin and eosin (H&E) and immunohistochemical staining of the colorectal adenoma, carcinoma and adjacent normal tissues from a single patient. (**B**) 34 tumor samples on tissue microarray were immunohistochemical stained and scored. Score distribution and average score of each antibody staining were summarized. (**C**) Scoring differences between tumor and matched normal tissues from twenty-six patients. * Denotes p-GSK3β$^{Y216}$, FAK or PYK2 scores in different directions. _ highlights the numbers of patients whose TMA scores of p-GSK3β$^{Y216}$, FAK or PYK2, and β-catenin move in the same direction. (**D**) Hematoxylin and eosin (H&E) and immunohistochemical staining of colorectal carcinoma and adjacent normal tissues from a familial adenomatous polyposis patient. (**E**) Schematic diagram of the FKA/PYK2/p-GSK3β$^{Y216}$/β-catenin axis and its role in intestinal tumorigenesis.

FAK/p-GSK3α/β$^{Y279/Y216}$/β-catenin (19 of 26 samples) and PYK2/p-GSK3α/β$^{Y279/Y216}$/β-catenin (18 out of 26 samples) were regulated in the same way, supporting that they are related events in the development of colonic cancer (*Figure 6C*). To definitively link the upregulation of FAK/PYK2/p-GSK3α/β$^{Y279/Y216}$/β-catenin axis in human CRC to the loss of functional APC, we analyzed serial sections of colon specimens from a patient with FAP, an inherited colorectal cancer syndrome caused by germline mutation of *APC*. The levels of FAK/PYK2, p-GSK3α/β$^{Y279/Y216}$ and β-catenin were all found substantially elevated in colon carcinomas (*Figure 6D*), thus providing proof-of-concept evidence linking these molecular changes to the loss of functional APC in CRC patients.

## Discussion

The current study has revealed a novel FAK/PYK2/GSK3β$^{Y216}$/β-catenin regulatory circuit in intestinal tumorigenesis. Our findings indicate that: loss of functional APC induces the expression of FAK and PYK2, elevated FAK/PYK2 phosphorylates GSK3β$^{Y216}$ to stabilize β-catenin, which initiates and supports intestinal tumorigenesis (*Figure 6E*). Phosphorylation of GSK3β$^{Y216}$ was discovered about two decades ago (*Hughes et al., 1993*), but until now, its role in Wnt/β-catenin signaling remains mysterious. Our results has uncovered a previously unrealized role of GSK3β$^{Y216}$ phosphorylation: it triggers GSK3β recruitment of β-TrCP and subsequent GSK3β monoubiquitination by β-TrCP, an event required for Wnt-induced inhibition of β-catenin degradation, thus identifying phosphorylation of GSK3β$^{Y216}$ as an 'activator' of the Wnt/β-catenin pathway.

Phosphorylation of GSK3β$^{Y216}$ has been shown to be required for GSK3β's full kinase activity measured by in vitro phosphorylation of c-jun (*Hughes et al., 1993*); however, mutation of Y216 did not impair GSK3β's ability to phosphorylate Tau (*Buescher and Phiel, 2010*). The reason for the discrepancy is not clear, but it might be explained by the fact that different substrates were used in these studies. The current study indicates that phosphorylation of GSK3β$^{Y216}$ per se does not affect GSK3β activity towards β-catenin, thus ruling out the possibility that GSK3β$^{Y216}$ regulates the Wnt/β-catenin pathway through modulating GSK3β kinase activity.

GSK3 is emerging as a prominent drug target for the treatment of various disorders including cancer and currently more than 50 GSK3 kinase inhibitors have been developed (*McCubrey et al., 2014*). The kinase activity per se of GSK3 plays a dual role in Wnt signaling: GSK3 phosphorylation of Wnt coreceptor LRP6 on its PPPSP motifs is required for the activation of the pathway (positive role), whereas phosphorylation of β-catenin by GSK3 triggers its degradation (negative role) (*Wu and Pan, 2010*). The kinase-mediated double role feature makes targeting GSK3 catalytic activity potentially problematic since it suppresses GSK3-mediated degradation of β-catenin, thus activating Wnt signaling therefore being potentially oncogenic. Indeed, 60 days of lithium treatment resulted in overall modest but significant increase in the tumor size in *APC*$^{min/+}$ mice (*Gould et al., 2003*), implying that GSK3 kinase inhibitor does pose an oncogenic risk, at least in the context of Wnt-driven carcinogenesis. The current study suggests a novel way of GSK3 inhibition: targeting GSK3α/β$^{Y279/Y216}$ phosphorylation, which opens a new avenue for the design and development of GSK3 inhibitors.

FAK plays a pivotal role in angiogenesis, tumor cell invasion and metastasis (*Sulzmaier et al., 2014*). FAK is also required for tumor initiation: organ-specific deletion of FAK suppresses mammary tumorigenesis (*Luo et al., 2009*; *Pylayeva et al., 2009*), intestinal tumorigenesis (*Ashton et al., 2010*), skin tumor formation (*McLean et al., 2004*) and hepatocarcinogenesis (*Shang et al., 2015*), suggesting that inhibition of FAK may present a promising strategy for treating these diseases. FAK has kinase-dependent and kinase-independent scaffolding functions (*Cance et al., 2013*); whether and how these two activities contribute to FAK's role in a particular type of cancer remains largely unexplored. In c-Met/β-catenin-driven hepatocarcinogenesis, FAK kinase activity was not required for β-catenin-induced Cyclin D1 expression (*Shang et al., 2014*). However, pharmacologic inhibition of FAK kinase activity suppressed KRAS-driven lung adenocarcinomas (*Konstantinidou et al., 2013*), suggesting an involvement of FAK kinase activity in KRAS-driven lung cancer. This study indicates that FAK contributes APC-driven intestinal tumorigenesis through its kinase activity. Currently several FAK inhibitors are undergoing clinical trials and the first phase I trial has been very promising. A clinical challenge that lies ahead is to identify tumor types and patient subgroups that are most responsive to FAK inhibitors. To this end, the current study has identified a potentially responsive tumor type—colorectal cancer—and potentially responsive patient subpopulations—CRC patients with APC mutations.

FAK is known to regulate β-catenin dynamics by directly phosphorylating β-catenin[Y142] (*Chen et al., 2012*). Phosphorylation of β-catenin[Y142] acts as a switch from the adhesive to the transcriptional role of β-catenin (*Piedra et al., 2003*); as a result, FAK facilitates vascular endothelial cadherin-β-catenin dissociation and endothelial cells junctional breakdown (*Chen et al., 2012*). This study has revealed a new role of FAK in the regulation of β-catenin: stabilizing β-catenin by phosphorylating GSK3α/β[Y279/Y216]. FAK regulates mammary stem cell and intestinal tissue regeneration (*Luo et al., 2009*; *Ashton et al., 2010*; *Sulzmaier et al., 2014*); given the importance of Wnt/β-catenin pathway in stem cell, our study could provide a potential mechanistic explanation how FAK is involved in stem cell and tissue regeneration.

PYK2 is closely related to FAK in sequence and structure with 48% amino acid identity, and they function redundantly and non-redundantly in a function and context-dependent manner (*Schaller, 2010*). Despite its low expression in normal human intestines (*Avraham et al., 1995*), we and others find that intestinal PYK2 is elevated in adenomas in $APC^{min/+}$ mice and in CRC patient tissues (*Carothers et al., 2006*). The current study clearly show that by phosphorylating GSK3α/β[Y279/Y216], FAK and PYK2 function redundantly in the regulation of Wnt/β-catenin signaling in CRC cells, raising the possibility that FAK and PYK2 could also function redundantly in Wnt-driven intestinal tumorigenesis. However, if so, why was intestine-specific deletion of FAK alone sufficient to cause complete abrogation of adenoma formation in APC-mutant mice (*Ashton et al., 2010*)? Why did PYK2 not compensate for the loss of FAK in intestinal tumorigenesis? A prior study (*Carothers et al., 2006*) and our data (*Figure 5C*, right panel) showed that drastic PYK2 elevation occurred in tumors in $APC^{min/+}$ mice, indicating that PYK2 is upregulated during the process of intestinal tumorigenesis. FAK induction, although further enhanced in tumors, however, is triggered by APC mutation—the initiating event for tumorigenesis in the APC-mutant mice, through activated Wnt pathway target c-Myc (*Ashton et al., 2010*); therefore FAK induction by APC mutation should occur earlier than PYK2 elevation. If so, it explains why inhibition of FAK by intestinal deletion of FAK was sufficient to suppress tumor development, it also explains why PF562271 treatment inhibited PYK2 elevation in $APC^{min/+}$ mice (*Figure 5C*, left panel). Nevertheless, regardless of the exact mechanisms mediating in vivo PYK2 regulation during intestinal tumorigenesis, elevated PYK2 in tumors further enhances Wnt-dependent tumorigenesis through phosphorylating GSK3 to promote the Wnt pathway—a positive feedback loop that reinforces the tumor development process.

Tumors undergoing targeted therapy often relapse due to the utilization of autonomous parallel-redundant signaling. The current study indicates that both FAK and PYK2 are elevated in already established colon tumors in CRC patients and that effective inhibition of the Wnt/β-catenin pathway in CRC cells expressing FAK and PYK2 will only be achieved by simultaneously targeting both kinases; we believe these results provide a solid mechanistic justification for clinical use of FAK/PYK2 dual inhibitor instead of FAK-specific inhibitor for better clinical outcomes in CRC patients.

# Materials and methods

## Statistical analysis

Data are presented as mean ±SD. The difference between two groups was evaluated using Student's *t*-test (two tailed). p values less than 0.05 were considered statistically significant.

## Additional methods

Detailed information for DNA constructs, shRNA, antibodies and reagents, and detailed methods for cell culture, immunoprecipitation (IP), immunoblotting (IB), transfection, lentivirus and retrovirus production and infection, in vitro kinase assay, ubiquitination assay, soft agar assay, cell proliferation assay, and animal experiments are provided below.

### Cell lines and cell culture

All cells were cultured in a 37°C humidified incubator containing 5% $CO_2$. HEK293T, SW480, and MEF cells were grown in DMEM. DLD-1 cells and HT-29 cells were cultured using RPMI-1640 medium and MoCoy's 5a medium, respectively. All cells were supplemented with 5% fetal bovine serum, 100 units/ml penicillin and 100 μg/ml streptomycin. Wnt3a-producing L cells and control L cells were obtained from ATCC and used for generating Wnt3a-conditioned medium (CM) and control-CM according to ATCC's

instruction. GSK3α$^{+/+}$ and GSK3β$^{-/-}$ MEFs were generously provided by James Woodgett (Ontario Cancer Institute, Canada). FAK$^{R454/R454}$ knockin MEFs were described previously (*Lim et al., 2010*).

## Plasmids

The expression plasmids of HA-GSK3β, GSK3α, FLAG-β-catenin and Myc-TrCP were obtained from Addgene (Cambridge, MA). Coding sequence for GSK3α was amplified and cloned into pcDNA3-FLAG to genetrate plasmid expressing FLAG-tagged GSK3α. Plasmids expressing ubiquitin- HA-GSK3β fusion protein and GSK3β ubiquitination mutant were previously described (*Gao et al., 2014*). GSK3β, GSK3α and β-catenin phosphorylation mutants were generated by PCR-directed mutagenesis. All the constructs are sequenced to verify the desired mutations. Lentiviral-based shRNA plasmids for targeting mouse GSK3α, human FAK and human PYK2 were purchased from Sigma (St Louis, MO). To construct lentiviral-based vectors for human FAK or mouse PYK2 knockdown (KD), primers containing the sequence of shRNA oligonucleotides for FAK (sense: 5′-GCCCAGGTTTACTGAACTTAA-3′) or PYK2 (sense: 5′-GAAGTAGTTCT-TAACCGCAT-3′) was annealed and ligated into pLKO.1-blasticidin which was obtained from Addgene. Lentiviral-based vector targeting human GSK3 (GSK3α/3β) was constructed by ligating annealed primers (5′-GAACCGAGAGCTCCAGATC-3′) into pLKO.1-puromycin (Addgene). Coding sequences for HA-tagged GSK3β, FLAG-tagged GSK3α and FLAG-tagged β-catenin in pcDNA3 were used as templates for cloning of these genes into pMX-IRES-BSR retroviral vector. shRNA-resistance was achieved by introducing synonymous mutations in the shRNA targeting sequence of plasmids using PCR-directed mutagenesis.

## In vitro ubiquitination assay

Wild type (WT) HA-GSK3β and HA-GSK3β$^{Y216}$ mutant proteins were synthesized using the TNT T7 quick-coupled transcription/translation kit (Promega, Madison, WI). To purify in vitro translated HA-GSK3β proteins, 5 μl out of 25 μl of the translation product were used for IP with EZview Red anti-HA affinity gel (Sigma). The immunoprecipitates were subjected to in vitro ubiquitination assay. The assay was carried out in 1× ubiquitination assay buffer (25 mM Tris-Cl at PH 7.5, 5 mM MgCl$_2$, 100 mM NaCl, 2 mM ATP, 1 mM dithiothreitol) at a final volume of 20 μl containing: 130 ng E1 (Boston Biochem, Cambridge, MA), 230 ng E2 UbcH5b (Boston Biochem), 300 ng SCF$^{βTrCP1}$ complex consisting of Rbx1, Cul1, Skp1 and β-TrCP1 (Millipore, Billerica, MA), 3 μg FLAG-Ubiquitin (Boston Biochem) and purified HA-GSK3β or its mutants. The reaction mixture was incubated at 37°C for 2 hr, and then subjected to SDS-PAGE followed by IB with anti-GSK3β antibody.

## In vitro kinase assay

The in vitro kinase assay to assess kinase activity of GSK3β and its mutants towards β-catenin was performed as previously described (*Gao et al., 2014*). To validate FAK/PYK2 as kinases phosphorylating GSK3β at Y216, recombinant FAK and PYK2 and in vitro translated HA-GSK3β were used in in vitro kinase assay. Briefly, in vitro translations were carried out using the rabbit reticulocyte Lysate TNT Coupled Transcription/Translation System (Promega) according to the protocol provided. Five out of 25 μl in vitro translated HA-GSK3β was diluted with lysis buffer and incubated with EZview Red anti-HA affinity gel (Sigma) at 4°C for 1 hr. The bead-bound immunoprecipitates were washed with lysis buffer containing 20 mM Tris-HCl, pH 7.5, 150 mM NaCl, 1 mM EDTA, 1% NP40 and 10% Glycerol for three times followed by a final kinase buffer wash, then incubated with 20 ng full length recombinant FAK (SignalChem, Canada) or PYK2 (Enzo, Farmingdale, NY), 2 mM ATP, kinase buffer (Cell Signaling, Danvers, MA) in a total volume of 15 μl at 37°C for 2 hrs. The reaction product was subjected to SDS-PAGE followed by IB with antibody recognizing p-GSK3β$^{Y216}$.

## IB, antibodies and reagents

Whole cell lysates were prepared in M-PER buffer (Thermo, Waltham, MA), resolved by SDS-PAGE and blotted with indicated antibodies. The following antibodies were used in this study: Anti-HA, anti-FLAG, anti-phospho-β-catenin (Ser33/37/Thr41), anti-β-catenin, anti-GSK3α, anti-β-TrCP, anti-α-Tubulin, anti-GSK3β, anti-FAK, anti-phospho-FAK (Tyr397), anti-PYK2 and anti-phospho-PYK2 (Tyr402) (Cell Signaling); anti-FLAG M2 (Sigma); anti-ubiquitin and anti-β-catenin (BD Bioscience, San Jose, CA); anti-GSK3, anti-c-Myc, anti-PYK2, anti-β-TrCP, anti-phospho-GSK3β (Tyr216) and anti-β-Actin (Santa Cruz Biotechnology, Dallas, TX); anti-phospho-GSK3 (Tyr279/Tyr216) (Millipore). Anti-GSK3β for IP was purchased from Abcam (Cambridge, MA) and Bethyl Laboratories (Montgomery, TX). SuperSignal West Pico Chemiluminescent Substrate and SuperSignal Western Blot Enhancer (Thermo) were used to enhance western signal when needed. PF-562271 was purchase from MedKoo Biosciences (Chapel Hill, NC) and Selleck Chemicals (Houston, TX). Gelucire is a gift from Gattefosse (Paramus, NJ). MG-132 was obtained from Sigma.

## IP

Briefly, cells were lysed with M-PER buffer or RIPA buffer (Thermo) supplemented with protease inhibitor and 20 mM N-ethylmaleimide (NEM) (Sigma). The whole cell lysates were pre-cleared with Protein G-sepharose beads at 4°C for 30 min. The cleared lysates were incubated with indicated antibody together with Protein G-sepharose beads or EZview Red anti-HA or anti-FLAG M2 affinity gel (Sigma) at 4°C for 4 hr. The immunoprecipitates were washed three times with lysis buffer containing 20 mM Tris-HCl, pH 7.5, 150 mM NaCl, 1 mM EDTA, 1% NP40 and 10% Glycerol, and subjected to SDS-PAGE followed by IB with indicated antibodies. Input corresponded to 5% of the total lysates used for the IPs and was verified by IB.

## Transient transfection, lentivirus production and infection

Plasmid transient transfections were performed using PolyJet In Vitro DNA Transfection Reagent (SignaGen, Rockville, MD) according to the manufacturer's instruction. Lentiviruses encoding the shRNA against target genes were produced in HEK293T cells by transfection of the lentiviral vector expressing shRNA against target gene with the third generation packaging systems (Addgene). The media containing viral particles were filtered through syringe filters and subsequently used to infect target cells. Cell lines stably expressing shRNA were established by selection. To generate FAK/PYK2 double KD cell line, DLD-1 cells were first infected with lentivirus particles encoding the shRNA against PYK2. PYK2 KD DLD-1 cell line was established by puromycin selection and used for second round infection with lentivirus particles encoding the FAK shRNA. Double KD cell line was obtained by puromycin and blasticidin selection. For retrovirus infection, the retroviral vectors expressing shRNA-resistant GSK3β or its mutants were transiently transfected into a potent retrovirus packaging cell line named platinum-A (Plat-A). 48 hr post transfection infectious retrovirus were collected and used to infect SW480 cells. The stable cell lines were established by blasticidin selection and used for cell proliferation assays. The cell lines overexpressing GSK3 shRNA-resistant GSK3β or its mutants were subjected to GSK3 KD by lentivirus-mediated RNA silencing. The resulting cell lines were used for anchorage-independent growth assays and xenograft experiments. The same strategy was used to generate SW480 cells expressing shRNA-resistant GSK3α or its mutants with endogenous GSK3 knocked down for xenograft experiments. HT-29 cells stably expressing FLAG-β-catenin$^{Y142E}$ were also generated by retrovirus-mediated gene transfer.

## Anchorage-independent growth assay

The SW480 cells were trypsinized and suspended in DMEM containing 0.4% low melting point (LMP) agarose (Promega). 2 ml of the above medium containing $3 \times 10^4$ cells was plated over the 3 ml bottom agar containing 0.8% LMP agarose in a 60 mm dish. Two independent experiments were performed in triplicates. The dishes were incubated at 37°C with medium replenished every 3 days until colony formed. The colonies were stained with 0.01% crystal violet solution. Pictures of representative field were taken under stereo microscope (Olympus SZX10).

## Cell proliferation assays

SW480 parental cells or cells overexpressing HA-tagged GSK3β or its phosphorylation mutants were used for cell proliferation assays. These cells were seeded into 96-well plates at a density of $5 \times 10^3$. 24 hr later, the medium was replaced by fresh medium containing 2 μm FAK/PYK2 inhibitor PF-562271. The cell viability was measured by CCK-8 kit (Sigma) according to manufacturer's protocol every day for 4 days. Three independent experiments were performed in triplicates.

## Animal experiments

All animal procedures were carried out according to protocols approved by the Institutional Animal Care and Use Committee at the University of Pittsburgh. Mice were fed a standard diet (Purina LabDiet, diet ID 5P75). For intestinal tumorigenesis experiments, age-matched female APC$^{min/+}$ mice on C57BL/6J background and C57BL/6J female mice (control mice) were obtained from the Jackson Laboratories. At 11 weeks of age, the APC$^{min/+}$ were randomized into two groups, receiving either vehicle (5% Gelucire) (n = 8) or PF-562271 (33 mg/kg in vehicle) (n = 8) by oral gavage twice daily for 5 weeks. After the mice were euthanized, the entire small intestinal tracts were carefully dissected, rinsed with ice-cold saline, opened longitudinally on ice. For polyp count experiments (n = 5 for vehicle- or PF-562271-treated group), the excised mice small intestine segments were fixed flat in 10% formalin for 48 hr. The fixed intestine was stained with 0.2% methylene blue (Sigma) for 2 min and then

destained in 75% ethanol to reduce background staining. Polyp number and polyp size were counted under a dissecting microscope. For the collection of mice intestine tissue (n = 3 for vehicle- or PF-562271-treated group, n = 2 for control C57BL/6J group), the mucosal layer of intestine (only freshly isolated intestines, not those already used for polyp counting) was harvested by gentle scraping with a glass slide and all procedures were performed on ice. After harvesting, the tissues were collected into tubes and immediately placed on dry ice. The tissue samples were kept a liquid nitrogen tank before processing for western blotting or IP analysis. For these analyses, the tissues were lysed on ice in RIPA buffer (Thermo) supplemented with phosphatase inhibitor (Thermo), protease inhibitor (Thermo) and 20 mM NEM (Sigma). For the indicated experiments, individual polyps in 5-month-old female $APC^{min/+}$ mice were isolated for tumor tissue lysates.

For xenograft experiments, $5 \times 10^6$ SW480 or HT-29 cells as indicated were injected subcutaneously into both flanks of 4- to 5-week-old athymic nude mice (Charles River), 5 mice in each group. The mice were housed in a sterile environment. Tumor size was determined by caliper measurements twice a week. The tumor volume was calculated using the formula: $V = \frac{1}{2} \times a \times b$ (*Sparks et al., 1998*), where a and b denoted the largest and smallest tumor axis, respectively. Mice were euthanized 24 days after implantation; tumors were excised, weighed and photographed. To test the efficacy of FAK/PYK2 inhibitor in xenograft model, 1 week of tumor injection, animals were treated with either vehicle (5% Gelucire) or PF-562271 (33 mg/kg in vehicle) by oral gavage twice daily for 3 weeks. Mice were euthanized 28 days after implantation.

## Immunohistochemistry

Formalin-fixed and paraffin-embedded tissue microarrays of human colonic cancer tissue microarray containing 34 cases of colorectal adenocarcinoma and 26 matched and 8 unmatched adjacent normal tissues were purchased from US Biomax Inc. The de-identified human colon tissue samples from a sporadic-colon-cancer patient and a familial adenomatous polyposis (FAP) patient, archived at the University Of Pittsburgh School Of Medicine, Department of Pathology, were obtained in compliance with a University of Pittsburgh Cancer Institute (UPCI) tissue banking protocol (UPCI 97-130). The immunohistochemical analysis was performed in compliance with the UPCI Institutional Review Board protocol, UPCI 08-026. Immunohistochemistry (IHC) was performed on 4-micron formalin-fixed paraffin-embedded tissue from either tissue microarray or colon cancer resection. Briefly, 4 µm paraffin sections were deparaffinized in xylene solutions and rehydrated in graded alcohol solutions followed by washes in distilled water. Antigen retrieval was performed in the pressure cooker for 15 min in 20 mmole/l Tris-EDTA buffer (pH 9.0). The sections were allowed to cool to room temperature and then incubated overnight in a humidified chamber at room temperature with indicated antibodies. After washing with PBS, the sections were incubated for 1 hr at room temperature with HRP-labeled polymer anti-mouse or anti-rabbit second antibody (DAKO Envision+ system, Carpinteria, CA), depending on the host which individual antibody was prepared. Color visualization was performed with liquid DAB chromogen in imidazole-HCl buffer (pH 7.5) containing hydrogen peroxide until the brown color fully developed. The sections were counterstained with hematoxylin and coverslippped with permanent mounting media. The intensity of TMA staining was score as 0 (negative), 1+ (weak), 2+ (moderate) and 3+ (strong). The following antibodies were used for immunohistochemical staining: anti-FAK (Millipore, Cat# 05-537, 1:100 dilution), anti-PYK2 (Bioworld, Cat# BS1420, 1:50 dilution), anti-GSK3β (Cell Signaling, Cat# 9315, 1:100 dilution), anti-phosphor-GSK3β (Tyr216) (Gene Tex, Cat# GTX38564, 1:100 dilution) and anti-β-catenin (Zymed, Cat# 18-0226, 1:200 dilution).

## Additional information

### Funding

| Funder | Grant reference | Author |
| --- | --- | --- |
| National Cancer Institute (NCI) | CA166197; CA175202 | Jing Hu |
| National Cancer Institute (NCI) | CA102310 | David D Schlaepfer |

The funder had no role in study design, data collection and interpretation, or the decision to submit the work for publication.

## Author contributions

CG, Conception and design, Acquisition of data, Analysis and interpretation of data, Drafting or revising the article; GC, Conception and design, Acquisition of data, Analysis and interpretation of data; S-FK, Acquisition of data, Analysis and interpretation of data; DHZ, Acquisition of data; DDS, Analysis and interpretation of data, Contributed unpublished essential data or reagents; JH, Conception and design, Analysis and interpretation of data, Drafting or revising the article

## Ethics

Animal experimentation: This study was performed in strict accordance with the recommendations in the Guide for the Care and Use of Laboratory Animals of the National Institutes of Health. All of the animals were handled according to approved institutional animal care and use committee (IACUC) protocols (#14013138) of the University of Pittsburgh.

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
