## [Decision Letter]

[Editors’ note: a previous version of this study was rejected after peer review, but the authors submitted for reconsideration. The previous decision letter after peer review is shown below.]

Thank you for choosing to send your work entitled “FAK/PYK2 Promotes the Wnt/β-catenin Pathway and Intestinal Tumorigenesis by Phosphorylating GSK3B” for consideration at *eLife*. Your full submission has been evaluated by Tony Hunter (Senior Editor) and two reviewers, one of whom is a member of our Board of Reviewing Editors. The decision was reached after discussions between the reviewers.

One reviewer had detailed comments that would be difficult to address in a timely way, and without which the significance of the work is unclear. The other reviewer had a general comment that the work is preliminary without more in vivo experiments. Based on our discussions and the individual reviews below, we regret to inform you that your work will not be considered further for publication in *eLife*.

Reviewer #1:

This paper provides a potential molecular mechanism for the observation that colorectal cancers (CRCs) frequently have high levels of FAK/PYK2 and increased nuclear β-catenin, and that FAK deletion suppresses adenomas in a mutant *APC* mouse model. This is potentially a second mechanism for FAK to promote β-catenin accumulation (in addition to the phosphorylation of β-catenin by FAK).

In a previous paper, the authors showed that Wnt stimulates monoubiquitination of GSK3 and this monoubiquitination reduces phospho-β-catenin presentation to β-TrCP, thereby stabilizing β-catenin. Here they find that phosphorylation of GSK3β^Y216^ stimulates monoubiquitination and binding of β-TrCP. The 216F mutant shows that lack of pY216 and/or lack of ubiquitin increases β-catenin turnover in presence or absence of Wnt, causing low β-catenin levels. The 216E mutant is constitutive. They show that FAK and PYK2 can phosphorylate Y216 (normally thought of as an autophosphorylation site), and knockdown of FAK reduces β-catenin level. FAK inhibitor (PF) inhibits tumor formation when GSK3 is wildtype, but only partly inhibits GSK3β^Y216E^ mutant tumors. However, the authors do not distinguish effects of pY216 from effects of ubiquitin by use of their KKKK mutant used in their JBC paper. Therefore, the observations could be due to reduced GSK3 auto or trans phosphorylation rather than ubiquitin. It is also remarkable that the low stoichiometry of ubiquitination has a measurable effect, because the non-ubiquitinated molecules should remain active. However, the results would be understandable if the subset of GSK3 that is ubiquitinated is in the destruction complex. Another surprising finding is that GSK3β^Y216^ is not needed to stimulate β-catenin degradation by β-TrCP, which is presumably requires phosphorylation of β-catenin by GSK3 – but they didn't measure phospho-β-catenin with the 216F mutant.

From the significance point of view, the authors introduce the paper by saying that 85% of CRCs have mutant *APC* yet only 50% have increased nuclear β-catenin, implying that *APC* mutation is not sufficient to up-regulate nuclear β-catenin. Their paper provides data in support of a mechanism in which increased FAK/PYK2 activity in CRCs leads to increased GSK3β^Y216^, increases GSK3 ubiquitination, inhibits β-catenin turnover by bTrCP, and raises nuclear β-catenin levels. This predicts that nuclear β-catenin would be detected in CRCs that lack *APC* mutations, rather than the converse. It isn't clear how many human CRCs have increased nuclear β-catenin but lack *APC* mutations. Putting that aside, the tumor growth studies with 216F mutant GSK3 support the importance of this site, and the cancer studies with the PF inhibitor and the pathology supports the importance of FAK/PYK2 in CRC. However, the inhibition of tumors by PF does not exclude GSK3-independent effects of FAK.

1) The KKKK mutant should be used to confirm that nuclear β-catenin is increased regardless of FAK and PYK2 activity.

2) KKKK mutation of Y216E mutant should suppress activation.

3) Phospho-β-catenin^S33/37^ should be measured with GSK3 YF, YE and KKKK mutants to confirm the inference that pY216 is not needed for phosphorylation of β-catenin.

4) The mechanism would be much better supported if the authors could show an effect of GSK3β^Y216^ or ubiquitination on the ability of β-TrCP to interact with phospho-β-catenin.

5) Does the PF inhibitor inhibit tumors in which the primary FAK phosphorylation site in β-catenin is mutated?

Reviewer #2:

The authors report a FAK/PYK2/GSK3β^Y216^/β-catenin signaling axis, through which FAK and PYK2 converge on GSK3β^Y216^ and GSK3α^Y279^, with phosphorylated GSK3β^Y216^ recruiting β-TrCP, which monoubituitinates GSK3β, blocking Wnt-induced degradation of β-catenin. All experiments were done in cell lines only, and a single FAK/PYK dual inhibitor was tested (although experiments were done in genetic and knockdown systems to verify signaling).

While experiments shown were convincing, the authors raise two issues not further addressed experimentally. First was that the dual FAK/PYK2 inhibitor also blocked GSK3α^Y279^, suggesting functional redundancy that was not further elucidated experimentally.

Second, the authors suggest but do not test, that FAK was upstream of PYK2 in vivo but not in vitro. Although the authors claim that use of a dual PYK2/FAK inhibitor might be superior to monospecific inhibitors to block emergent resistance in patients, the relevance of their in vitro experiments to an in vivo setting remain an open question not addressed in experiments presented. In the absence of in vivo validation, the conclusions drawn remain fairly preliminary.

---

## [Author Response]

[Editors’ note: the author responses to the previous round of peer review follow.]

We have made every effort to address the critiques thoroughly in this revision; the results of our additional experiments lend further support for our conclusion. Our response to the reviewers’ specific comments and changes made during the revision are detailed below.

Reviewer #1:

1) The KKKK mutant should be used to confirm that nuclear β-catenin is increased regardless of FAK and PYK2 activity.

As suggested, we now included results from the experiments using the KKKK15/27/32/36 RRRR mutant (S4) and the S4-ubiquitin fusion mutant (Ub-S4, mimicking ubiquitinated form of S4 mutant). As shown in Figure 4 (lanes 7-10), in R454/PYK2 KD MEFs with deficient FAK and PYK2 activity, ectopic expression of Ub-S4 mutant protein, but not S4, restored Wnt-induced β-catenin, confirming the role of GSK3β ubiquitination in the activation of Wnt signaling.

*2) KKKK mutation of Y216E mutant should suppress activation*.

As predicted by the reviewer, our new data (Figure 2, lanes 13 and 14) showed that reconstitution of GSK3β KO/GSK3α KD MEFs with GSK3β^Y216E/S4^ mutant (Y216 phosphorylation mimetic lacking ubiquitination) inhibited Wnt-induced β-catenin accumulation.

Together, our new results (Figure 4, lanes 7-10; and Figure 2 lanes 13 and 14) confirmed that it is phosphorylation-dependent *ubiquitination* that mediates β-catenin stabilization upon Wnt stimulation.

*3) Phospho-β-catenin*^*S33/37*^
*should be measured with GSK3 YF, YE and KKKK mutants to confirm the inference that pY216 is not needed for phosphorylation of β-catenin*.

As suggested, we have now included new results showing that mutation of GSK3β tyrosine 216 (to either phenylalanine or glutamic acid) or lysines 15/27/32/36 to arginines did not affect GSK3β activity towards β-catenin phosphorylation at serine S33 and 37 in vitro (Figure 1, upper panel) and in cells (Figure 1, lower panel).

*4) The mechanism would be much better supported if the authors could show an effect of GSK3β*^*Y216*^
*or ubiquitination on the ability of β-TrCP to interact with p-β-catenin*.

We conducted new experiments to address this issue. The new results (Figure 2, right panel) showed that upon Wnt3 stimulation, β-TrCP-bound p-β-catenin^S33/S37/T41^ was substantially more in GSK3β KO/GSK3α KD MEFs reconstituted with HA-GSK3β^Y216F^ than in MEFs reconstituted with WT HA-GSK3β or HA-GSK3β^Y216E^, implying that phosphorylation status of GSK3β^Y216^ impacts phosphorylated β-catenin^S33/S37/T41^ recruitment of β-TrCP_._

5) Does the PF inhibitor inhibit tumors in which the primary FAK phosphorylation site in β-catenin is mutated?

We have conducted a cell line xenograft study to address this important question, and the answer is yes. Our new data (Figure 4) showed that although overexpression of β-catenin^Y142E^ partially jeopardized the anti-tumor effect of PF562271, PF562271 treatment significantly inhibited in vivo growth of SW480 cells ectopically expressing Flag-β-catenin^Y142E^. While this result further proved the importance of FAK/PYK2-mediated GSK3 phosphorylation, it also suggested that phosphorylation of β-catenin^Y142E^ plays a role in FAK-targeted therapy for CRC.

Reviewer #2:

*[…] While experiments shown were convincing, the authors raise two issues not further addressed experimentally. First was that the dual FAK/PYK2 inhibitor also blocked GSK3α*^*Y279*^*, suggesting functional redundancy that was not further elucidated experimentally*.

We have performed new experiments to address the role of GSK3α^Y279^; (1) in the activation of Wnt/β-catenin pathway and (2) in vivo growth of SW480 cells. Our new results showed that Wnt-induced β-catenin accumulation was inhibited in GSK3β KO/GSK3α KD MEFs reconstituted with GSK3α^Y279F^ but not with GSK3α^Y279E^ (Figure 2, comparing lanes 9-10 with 11-12), hence validating the role of GSK3α^Y279^ phosphorylation in the activation of Wnt signaling. Our new data also showed that in GSK3α/β KD SW480 cells, reconstitution of the cells with WT HA-GSK3α or HA-GSK3α^Y279E^ but not HA-GSK3α^Y279F^ restored cell growth in a xenograft assay (Figure 2), implying the functional importance of GSK3α^Y279^ phosphorylation in CRC cells.

*Second, the authors suggest but do not test, that FAK was upstream of PYK2 in vivo but not in vitro. Although the authors claim that use of a dual PYK2/FAK inhibitor might be superior to monospecific inhibitors to block emergent resistance in patients, the relevance of their in vitro experiments to an in vivo setting remain an open question, not addressed in experiments presented. In the absence of in vivo validation, the conclusions drawn remain fairly preliminary*.

We thank the reviewer for raising this important issue. The fact that PYK2 activity towards GSK3 phosphorylation was intact in FAK^R454/R454^ knockin MEFs (Figure 4) indeed excluded FAK as direct upstream of PYK2 activity in vitro. We also recognized that it was not an accurate statement that “PYK2 and FAK function redundantly in intestinal tumorigenesis.” In the revised manuscript, we propose a new scenario to explain why intestine-specific deletion of FAK alone was sufficient to block tumor development in *APC* mutated mice.

A prior study (9) and our new data (Figure 5, right panel) showed that drastic elevation of PYK2 occurred in tumors in *APC*^min/+^ mice, indicating that PYK2 is upregulated during the process of intestinal tumorigenesis. FAK induction, although further enhanced in tumors, however, is triggered by *APC* mutation – the initiating event for tumorigenesis in the *APC*-mutant mice, through activated Wnt pathway target c-Myc (1). Therefore FAK induction by *APC* mutation should occur earlier than PYK2 elevation. If so, it explains why inhibition of FAK by intestinal deletion of FAK was sufficient to suppress tumor development. It also explains why PF562271 treatment inhibited PYK2 elevation in *APC*^min/+^ mice (Figure 5, left panel). By phosphorylating GSK3 to promote the Wnt pathway, elevated PYK2 in tumors further enhances Wnt-dependent tumorigenesis – a positive feedback loop that reinforces the tumor development process (as suggested in [9]). In the revised manuscript, corresponding changes have been made to the Discussion.

Regardless of the exact mechanisms mediating in vivo PYK2 regulation during intestinal tumorigenesis, both FAK and PYK2 are elevated in already established colon tumors in CRC patients (Figure 6). The current study indicates that effective inhibition of the Wnt/β-catenin pathway in CRC cells expressing FAK and PYK2 will only be achieved by simultaneously targeting both kinases, which provides a solid mechanistic justification for clinical use of FAK/PYK2 dual inhibitor instead of FAK-specific inhibitor for better clinical outcomes in CRC patients.